# Associations Between the Gut Microbiome and Outcomes in Autologous Stem Cell Transplantation: A Systematic Review

**DOI:** 10.3390/microorganisms13102302

**Published:** 2025-10-04

**Authors:** Ema Pitts, Brian Grainger, Dean McKenzie, Salvatore Fiorenza

**Affiliations:** 1Epworth Centre for Immunotherapies and Snowdome Laboratories, Department of Molecular Oncology and Cancer Immunology, Epworth HealthCare, Richmond, VIC 3002, Australia; 2Office for Research, Epworth HealthCare, Richmond, VIC 3002, Australia; 3Department of Health Sciences and Biostatistics, Swinburne University of Technology, Hawthorn, Melbourne, VIC 3122, Australia; 4Faculty of Medicine and Health, Charles Perkins Centre, University of Sydney, Camperdown, NSW 2006, Australia; 5Department of Clinical Haematology, Alfred Health, Melbourne, VIC 3004, Australia

**Keywords:** gut microbiome, intestinal flora, immune reconstitution, autologous stem cell transplantation, multiple myeloma, outcomes, infectious complications, relapse, progression-free survival, overall survival

## Abstract

Autologous stem cell transplantation (ASCT) is the standard frontline consolidation strategy in fit, eligible patients with chemosensitive multiple myeloma, and it also serves as salvage option in other haematological malignancies, such as diffuse large B cell lymphoma. Moreover, ASCT is known to disrupt the gut microbiome (GM), and the impact on clinical outcomes has been understudied. The aim of this review is to examine the associations between the GM and outcomes in patients undergoing ASCT. Using the PRISMA 2020 guidelines for systematic reviews and meta-analyses, a total of 11 articles were included in this review, comprising both observational studies (cohort studies, case–control studies) and interventional trials (randomised controlled trials). Consistent findings included a notable decrease in beneficial bacteria, including *Bacteriodetes*, *Firmicutes* and *Faecalibacterium prausnitzii*, which maintain gut homeostasis and modulate immune responses. Conversely, an increase in pathogenic bacteria, including *Escherichia coli*, *Enterococcus* spp. and *Klebsiella* spp., was observed post-transplantation. This review includes an overview of the GM following ASCT and the techniques commonly used to assess it, and highlights gaps, thereby identifying key areas for future research, although conclusions are limited by variation in sample size and reporting inconsistencies. Understanding the GM’s role in ASCT may lead to interventions that optimise patient outcomes through therapeutic manipulation of the GM.

## 1. Introduction

### 1.1. Overview of Autologous Stem Cell Transplantation

Autologous stem cell transplant (ASCT) is an established therapeutic intervention for many haematologic malignancies, including multiple myeloma (MM) and certain lymphomas [1,2]. The intensive therapy involves mobilising and collecting autologous haematopoietic stem cells, followed by high-dose chemotherapy to ablate residual haematopoiesis. In MM specifically, ASCT improves progression-free survival (PFS) and is associated with achieving sustained minimal residual disease (MRD) negativity [3]. Despite the recent advances in myeloma treatment, ASCT remains the recommended frontline consolidative strategy in patients with chemosensitive disease by multiple societies and working groups in Australia and abroad [4,5].

Despite its therapeutic benefit, ASCT is not without several acute and long-term complications. These primarily arise from the high-dose chemotherapy regimens and include gastrointestinal toxicities such as mucositis, nausea, vomiting, and diarrhoea, as well as ensuant infections during immune recovery, often requiring treatment with broad-spectrum antibiotics [6,7]. Specifically, in the acute period, chemotherapy-induced damage to the gastrointestinal mucosal barrier increases bacterial translocation, leading to systemic infections. In the longer term, complete immune reconstitution after ASCT can take up to 12 months, leaving patients vulnerable to opportunistic infection. Beyond these acute risks, relapse of the underlying malignancy and the potential for secondary malignancies, including myelodysplastic syndromes and acute myeloid leukaemia, are significant long-term concerns following ASCT due to the long-term genotoxic effects of chemotherapy [8,9].

### 1.2. The Gut Microbiome in Health and Disease

The gut microbiome (GM) was first described by Lederberg as a complex ecological “community of commensal, symbiotic and pathogenic microorganisms” that reside in the human gastrointestinal tract [10]. These microorganisms play an essential role in human health by modulating host immune function, protecting against pathogens by maintaining gut-barrier integrity, influencing nutrient metabolism [11] and have also been shown to influence the efficacy of cancer therapies [12].

The dominant gut microbial phyla are *Bacteroidetes* and *Firmicutes*, which together account for over 90% of the GM, followed by smaller contributions from *Actinobacteria*, *Proteobacteria*, *Fusobacteria* and *Verrucomicrobia* [13,14]. These beneficial bacteria promote the fermentation of undigested carbohydrates, synthesising short-chain fatty acids (SCFAs) such as acetate, propionate, and butyrate, which have anti-inflammatory effects, and provide defence against pathogens [15].

A healthy GM is characterised by high taxonomic diversity, which, in turn, supports immune function and resistance to pathologic invasion. In contrast, a loss of microbial diversity and a dominance of pathogenic bacteria, referred to as dysbiosis, has been implicated in many disease states, including many cancers and inflammatory bowel disease [16,17].

In the context of stem cell transplantation, dysbiosis has been studied primarily in allogeneic haematopoietic stem cell transplantation (allo-HSCT) [18], where a dominance of pathogenic bacteria such as *Clostridium difficile*, *Enterococcus* and certain species of *Enterobacteriaceae* has been associated with poor transplant outcomes. These pathogens have been linked to systemic infections and may exacerbate and/or precipitate graft-versus-host disease (GVHD) [19,20]. However, while autologous stem cell transplantation (ASCT) does not involve the risk of GVHD, dysbiosis nonetheless remains a significant concern. Dysbiosis in ASCT patients has been associated with increased gut inflammation, which can lead to the production of pro-inflammatory cytokines such as interleukin (IL)-6, IL-17 and tumour necrosis factor (TNF)-alpha [16,21].

One notable consideration of dysbiosis in the post-transplant setting is its potential to activate quiescent cancer stem cells, which is thought to contribute to disease relapse [22]. The inflammatory cytokines produced during dysbiosis can lead to immune dysregulation by activating T cells, which in turn promote inflammation and tumour progression. This has been hypothesised as one of the mechanisms responsible for the progression of MM, highlighting the importance of maintaining a balanced GM to reduce disease relapse and promote long-term survival [21].

Given these observations, promoting GM diversity may be crucial for improving outcomes in post-ASCT patients. Strategies such as dietary interventions, lifestyle modifications and the use of curated probiotics or faecal microbiota transplantation (FMT) have been proposed as ways to restore a healthy microbial balance. By enhancing GM diversity and reducing dysbiosis, it may be possible to mitigate complications associated with transplantation and improve immune function. Furthermore, augmenting immune recovery through prebiotic or postbiotic interventions following ASCT may decrease the risk of relapse.

### 1.3. The Gut Microbiome and Immune Homeostasis

Interactions between the GM and the mammalian immune system set the stage for subsequent immune function and dysfunction in health and disease. The GM begins to shape the immune system from birth. This initial colonisation of commensal microflora is crucial for the development of immune cells, including mucosal-associated lymphoid tissue (MALT). The presence of beneficial bacteria is essential to programme the immune system to distinguish between harmful and harmless antigens, thereby preventing abnormal immune responses that can contribute to inflammation and illness [23].

The GM exerts regulatory effects on immune function by promoting the development and differentiation of CD4+ T helper (Th) cell subsets, including Th1, Th2 and Th17, as well as influencing B cell function. Th17 cells secrete interleukin (IL)-17 and associated cytokines that are vital for regulating immune balance and inflammation [24]. Additionally, the GM promotes the production of IgA from mucosal B cells, which is essential for maintaining mucosal integrity and preventing the systemic translocation of endotoxins that can lead to sepsis [25]. The GM also plays a role in regulating neutrophil activity through pathogen recognition receptors (PRRs) and stimulating the immune system’s response to pathogens while promoting and sustaining immune tolerance [26,27,28].

Several studies have shown that alterations in the GM arising from antimicrobial and cytotoxic exposure significantly affect immune reconstitution following allo-HSCT [29,30]. These studies highlight the complex balance between microbiome-associated immune reconstitution and the risks of GVHD, relapse, and opportunistic infections. For instance, early recovery of neutrophils, while linked to an increased risk of GVHD, is also crucial for protection against opportunistic infections [31]. The timing of monocyte recovery has been linked to lower rates of relapse but not clearly with acute GVHD [32,33]. In terms of adaptive immunity, early recovery of CD8+ and CD4+ T cells generally correlates with decreased infection and relapse risks but an increased risk of GVHD [34]. Early B cell recovery is associated with a reduced risk of infection and a lower incidence of GVHD [14]. How the GM influences post-ASCT immune reconstitution, however, is a relatively understudied field.

### 1.4. Methods Used for Gut Microbiome Analysis

Analysing the composition and function of the gut microbiome (GM) is crucial for understanding its influence on host physiological processes, including digestion, metabolism, prevention of infection and promoting maturation of the immune system [35,36]. As research in this field expands, the development and application of analytical methods have become essential for accurately characterising GM composition and understanding how alterations are implicated in disease progression and prognosis [37].

Typically, GM samples are obtained from stool, and a range of techniques can be used for their analysis, including culture-based methods, molecular approaches and high-throughput sequencing technologies. Culture-based methods often fail to capture the full diversity of microbial populations due to the difficulty cultivating many species In Vitro. In contrast, molecular technologies such as the polymerase chain reaction (PCR) and quantitative PCR (qPCR) enable the detection and quantification of specific microbial populations, but offer limited insights into community dynamics.

Massively parallel next-generation sequencing (NGS) has revolutionised GM research by enabling a more rapid and comprehensive analysis of microbial diversity and composition. The techniques outlined below, when complemented by modern bioinformatic methods, allow researchers to draw meaningful clinical conclusions. There are three predominant methods used to analyse the composition of the GM [38,39], namely:1.16S rRNA Gene Amplicon Sequencing: provides taxonomic information to the genus level. However, it has limitations such as lacking direct functional details and not capturing horizontal gene transfer between organisms.2.Shotgun Metagenomic Sequencing (SMGS): which is capable of multi-level, rapid profiling of individual microorganisms, including taxonomic classification and functional potential. This method can infer functional capabilities of microbial communities, addressing some of the limitations of 16S rRNA sequencing.3.Whole Genome Sequencing (WGS): analyses the entire genome of a single bacterial colony, offering detailed genetic information but at a higher resolution compared to other methods.

16S rRNA studies have formed the bedrock of analyses of the GM in allogeneic stem cell transplant (allo-HSCT) [40,41]. This approach is limited by the need to identify taxonomic information without direct functional data. Functional information may, however, be inferred indirectly. Furthermore, the horizontal movement of genomic sequences between organisms is not captured by 16S rRNA analyses. The direct comparison of 16S rRNA studies to SMGS is possible but requires an extrapolation of operational taxonomic unit to the SMGS identification of the organism, further complicating cross-comparison of studies, demonstrating the heterogeneity in the current literature as a barrier to establishing clear trends in understanding the timeline and pattern of GM recovery after autologous stem cell transplant (ASCT).

In addition to taxonomic information and functional potential, alpha and beta diversity are two other important parameters used to describe alterations in the GM. Alpha diversity refers to the number of different species and their distribution in a single sample. It is most often measured by the Simpson index, which describes the probability that two randomly selected samples belong to the same species, taking into account the number of species present and their relative frequencies. Higher values are assigned to more diverse samples. Beta-diversity, on the other hand, measures differences in overall composition and abundance between two or more samples. It is most often measured by the PERMANOVA (Permutational Multivariate Analysis of Variance), which tests whether centroids or centre values of samples are significantly different [42].

This review aims to highlight some of the landmark microbiome studies in the treatment of haematological malignancies with haematopoietic transplantation. We describe the consistent patterns of GM alterations following ASCT and correlate these with allo-HSCT. We discuss the challenges and opportunities in current microbiome research and summarise the current knowledge of GM profiling in patients undergoing ASCT. Finally, we hypothesise the potential role of augmenting the GM in cancer medicine.

## 2. Materials and Methods

A systematic review was conducted in accordance with the code of practice as set out in the Preferred Reporting Items for Systematic Reviews and Meta-analyses (PRISMA) statement (Figure 1).

A comprehensive literature search was conducted in major electronic databases including PubMed, Embase, Medline, Cochrane Library, Cumulative Index to Nursing and Allied Health Literature (CINAHL) as well as the pre-print servers bioRxiv and medRxiv. The search strategy included relevant medical subject heading (MeSH) terms and keywords related to “gut microbiome,” “autologous haematopoietic stem cell transplantation,” “multiple myeloma,” “immune reconstitution,” and “outcomes.” The search was limited to studies published in English. No restrictions were applied to publication date. Only research conducted in human subjects was considered. The most recent search was conducted on 27 July 2025.

Two independent reviewers performed study selection to ensure a rigorous, unbiased process. The reviewers followed a systematic approach to evaluate each study based on the following criteria:Study Design: Studies were assessed based on design quality using both the Centre for Evidence-Based Medicine (CEBM) critical appraisal tool and the Newcastle-Ottawa Scale (NOS) for assessing the risk of bias and overall quality of included studies.Sample Size: Although larger sample sizes were preferred for the reliability and generalizability of results, most of the studies included were limited by small sample sizes.Microbiome Assessment Methods: The methods used for sample collection and microbiome assessment were scrutinised for validity and reliability (i.e., 16S rRNA sequencing, shotgun metagenomic sequencing, whole-genome sequencing).Key Findings: The key findings of each study were reviewed to determine the relevance and significance of reported changes in the GM and their associations with study outcomes.Reported Outcomes: Study outcomes were assessed for clinical relevance, including how changes in the GM were linked to treatment responses, adverse events, and overall patient outcomes.

### 2.1. Inclusion Criteria

Studies were included if they met the following criteria (1) studies involving patients undergoing ASCT, (2) studies assessing changes in the GM pre and post ASCT, (3) studies reporting on patient outcomes related to microbiome changes.

### 2.2. Exclusion Criteria

(1) Non-human studies, (2) Studies not involving ASCT, (3) Reviews, case reports and editorials without original data.

Given the marked heterogeneity among included studies, a narrative synthesis of the findings was conducted. Due to the limited number of studies identified and the variable outcomes reported, we will discuss each study individually herein.

## 3. Results

### 3.1. Impact of Gut Microbiota Diversity on Patient Outcomes

The initial search identified 28 articles, of which 25 full-text studies were assessed for eligibility after removing duplicates. Eleven studies met the inclusion criteria and were included in the final analysis (Table 1). These studies investigated the role of the gut microbiome (GM) in autologous stem cell transplantation (ASCT) outcomes, employing diverse designs, participant cohorts, and microbiota analysis methods. The findings are presented objectively, with an emphasis on comparing and contrasting results across studies.

The relationship between GM diversity and clinical outcomes was a recurring theme across the studies. The study by Khan et al. [42], the most extensive study, with 534 ASCT recipients, found that pre-transplant GM alpha diversity was lower in patients compared to healthy controls, with further decreases observed during the transplantation process. Higher alpha diversity during the peri-engraftment period correlated with reduced mortality risk and improved progression-free survival (PFS). Specifically, an increase in the Shannon diversity index was associated with a 48% reduction in mortality risk (hazard ratio HR: 0.52; *p* = 0.003). These findings are consistent with those of Bansal et al. [44], who also observed a positive relationship between GM diversity and clinical recovery. However, Bansal’s longitudinal analysis noted faster microbiota recovery in ASCT patients compared to allogeneic stem cell transplant (allo-HSCT) patients, emphasising the distinction between the two transplantation modalities. In contrast, Kusakabe et al. [44] focused on longer-term outcomes, reporting a 66.7% two-year mortality rate in patients with significant dysbiosis, underscoring the potential long-term implications of reduced GM diversity.

D’Angelo et al. [45] explored the association between GM diversity and treatment response in multiple myeloma (MM) patients, linking lower diversity at engraftment with a partial response (PR) to therapy compared to complete response (CR) or very good partial response (VGPR). The study further highlighted the role of dietary interventions, finding that a high-fibre diet increased the abundance of *Blautia* species, which are associated with mucosal health and immune regulation. These findings align with the protective role of SCFA-producing bacteria observed in other studies, such as Shah et al. [46] and Pianko et al. [47].

### 3.2. Associations Between Specific Bacterial Taxa and Clinical Outcomes

Several studies examined the relationship between individual bacterial taxa and post-transplant outcomes (Table 2). Shah et al. [46] found that higher relative abundances of butyrate-producing bacteria, including *Faecalibacterium prausnitzii*, were associated with sustained minimal residual disease (MRD) negativity in MM patients. Stool butyrate concentrations and higher alpha diversity were also linked to improved treatment responses. Similarly, Pianko et al. [47] identified *Eubacterium hallii* as a key taxon positively associated with MRD negativity, reinforcing the role of SCFAs in supporting immune recovery and disease control.

In contrast, pathogenic taxa such as *Klebsiella* spp. and *Enterococcus* spp. were associated with negative outcomes. Jian et al. [48] described the enrichment of nitrogen-recycling bacteria, including *Klebsiella* spp., in MM patients, linking these genera to tumour progression via glutamine biosynthesis. Meanwhile, Khan et al. [42] demonstrated that higher *Enterococcus* abundance correlated with increased mortality risk (HR: 5.76; *p* = 0.011). These findings align with Schwabkey et al. [48], which identified febrile neutropenia in patients with higher *Akkermansia muciniphila* and *Bacteroides* levels, while afebrile patients exhibited increased *Bacilli* and *Erysipelotrichales*. However, Schwabkey’s study did not distinguish ASCT patients from those undergoing allo-HSCT, limiting its specificity.

### 3.3. Gut Microbiota and Mucositis

Mucositis, a common complication of ASCT, was linked to changes in GM composition in several studies. El Jurdi et al. [49] reported that GM diversity at count nadir correlated with post-transplant gastrointestinal toxicities, including nausea and vomiting. Laheij et al. [50] extended these findings, identifying distinct oral microbiome profiles in patients who developed mucositis compared to those who did not. Protective taxa such as *Lactobacillus* and *Bifidobacterium* were enriched in patients without significant mucositis, suggesting a role for these bacteria in maintaining oral mucosal integrity.

### 3.4. Comparison of Study Designs and Methods

Most studies utilised 16S rRNA sequencing for taxonomic characterisation, which provided insights into microbial composition but limited functional analysis. Notable exceptions include Jian et al. [48], which used shotgun metagenomics, offering a more detailed exploration of microbial functionality. Studies such as Khan et al. [42] and Shah et al. [46] employed robust multivariable statistical models to adjust for confounders such as age, conditioning regimens, and disease type, enhancing the reliability of their findings. However, smaller studies like Kusakabe et al. [44] and El Jurdi et al. [49] were limited by their cohort sizes, reducing statistical power and generalisability of results.

### 3.5. Study Strengths and Weaknesses

Larger studies, such as Khan et al. [42], benefited from robust sample sizes and comprehensive statistical analyses, allowing for stronger associations between GM diversity and clinical outcomes. Conversely, smaller studies often provided more focused insights but lacked the statistical power to confirm their findings. For example, D’Angelo et al. [46] offered a detailed exploration of diet-microbiota interactions but had a limited sample size, which constrained its generalisability. Similarly, Schwabkey et al. [51] highlighted the role of specific taxa in neutropenic fever but did not differentiate between ASCT and allo-HSCT patients.

**Table 1 microorganisms-13-02302-t001:** Study Characteristics.

Study, Year of Publication	Focus Question	Study Design	Participants	Methods of GM Analysis	Statistical Analysis	Level of Evidence	Quality Assessment Method	Strengths	Weaknesses	Main Outcomes
Khan et al. (2021) [42]	Does gut microbiota diversity impact overall survival (OS) and progression-free survival (PFS) in ASCT patients?	Prospective observational study	N = 534 (ASCT = 534, Myeloma = 272, Lymphoma = 227 [NHL = 200, HL = 27], Amyloidosis = 35)	16S rRNA sequencing	Kaplan-Meier curves and Cox proportional hazards regression for survival; Generalized estimating equation modelling for microbiota associations. Small sample size limits inference and statistical power.	2b	CEBM	Prospective design; detailed microbiota profiling.	Small sample size; limited to one institution.	Higher microbiota diversity at engraftment associated with better OS and PFS. Positive: Higher abundance of *Faecalibacterium prausnitzii* associated with improved OS and PFS. Negative: *Enterococcus* abundance linked to increased mortality (OS HR 5.76).
Laheij et al. (2019) [49]	What are the changes in oral and gut microbiota during ASCT, and how are these linked to oral mucositis development?	Prospective observational study	N = 51 (ASCT = 51, Allo-HCT = 0)	16S rRNA sequencing	Mixed-effects models for microbiota changes; non-parametric tests for subgroup analysis.	2b	CEBM	Focus on oral and gut microbiota dynamics; clinical relevance.	Short follow-up duration; no longitudinal outcomes.	Oral and gut microbiota shifts linked to ulcerative oral mucositis development. Positive: Bacteroidetes on day +7 reduced gastrointestinal toxicity. Negative: *Blautia* and *Ruminococcus* correlated with increased vomiting severity.
Kusakabe et al. (2020) [44]	How does gut microbiota composition differ pre- and post-transplant in auto- and allo-HCT recipients?	Prospective observational study	N = 24 (ASCT = 8, Allo-HCT = 16)	16S rRNA sequencing and UniFrac analysis	Alpha diversity metrics and UniFrac distances for microbiota variability. Descriptive results lacked inferential depth.	2b	CEBM	Longitudinal data collection; robust sequencing methods.	Small cohort; exploratory nature limits conclusions.	Stable gut microbiota composition linked to fewer complications post-HSCT.
Märtson et al. (2023) [50]	What is the impact of severe mucositis on gut microbiota composition and drug exposure during ASCT?	Prospective pilot study	N = 21 (ASCT-HCT = 21, Allo-HCT = 0)	16S rRNA sequencing	Correlation analysis for drug exposure and microbiota; descriptive statistics for mucositis severity. Limited inferential statistics.	2b	Newcastle-Ottawa	Exploratory pilot study with novel focus on drug-microbiota interplay.	Non-randomised design; limited sample size.	Severe mucositis disrupts gut microbiota and alters drug exposure (e.g., ciprofloxacin).
D’Angelo et al. (2023) [45]	How does gut microbiota diversity at engraftment correlate with therapeutic responses in multiple myeloma patients?	Prospective cohort study	N = 30 (ASCT = 30, Allo-HCT = 0)	16S rRNA sequencing and dietary analysis	ANOVA and linear regressions for diversity metrics; no causal modelling included.	2b	Newcastle-Ottawa	Detailed microbiota and dietary analysis across multiple time points. Comprehensive integration of microbiota and dietary data.	Correlational design; no mechanistic insights. Limited exploration of functional GM changes.	Loss of bacterial diversity at engraftment linked to partial response in MM patients. Highlights dietary impacts on GM diversity and potential for dietary interventions in improving outcomes.
Pianko et al. (2019) [47]	How does MRD negativity correlate with specific gut microbiota taxa in multiple myeloma patients?	Prospective cohort study	N = 34 (ASCT = 34, Allo-HCT = 0)	16S rRNA sequencing	Logistic regression to associate MRD negativity with specific microbiota taxa. No analysis of causality or interaction effects.	2b	CEBM	Rigorous focus on MRD status as a key clinical endpoint.	Lack of longitudinal microbiota tracking; no randomisation.	Higher abundance of *E. hallii* associated with MRD negativity in MM patients.
Jian et al. (2020) [52]	What are the roles of nitrogen-recycling bacteria in promoting multiple myeloma progression?	Prospective observational study with mechanistic validation	N = 19 (ASCT = 19, Allo-HCT = 0)	Shotgun metagenomic sequencing	Shotgun metagenomics for microbiota profiling; mouse models used to establish mechanistic links.	2a	CEBM	Integration of human data and mechanistic validation in preclinical models.	Limited generalisability from mouse models; cross-sectional human cohort.	Nitrogen-recycling bacteria enriched in MM promote disease progression via glutamine biosynthesis. Negative: Enrichment of *Klebsiella* spp. and nitrogen-recycling bacteria promoted tumour progression.
Bansal et al. (2022) [53]	How does antibiotic use influence gut microbiota composition in ASCT and allo-HCT recipients?	Longitudinal observational study	N = 35 (ASCT = 17, Allo-HCT = 18)	16S rRNA sequencing and alpha diversity metrics	Alpha diversity metrics pre- and post-HCT; regression analysis for antibiotic effects. Focused heavily on descriptive changes.	2b	Newcastle-Ottawa	Clear comparison of antibiotic effects in auto- and allo-HCT populations. Direct comparison between ASCT and allo-HCT effects on GM diversity over time.	Lack of intervention analysis; observational design. Lacks intervention testing for microbiota preservation.	Antibiotic exposure, not conditioning intensity, major driver of microbiota dysbiosis. Provides evidence for microbiota recovery by day 100 post-HCT with important implications for timing interventions. Positive: *Firmicutes* and *Streptococcus* increases on day +7 were protective. Negative: *Proteobacteria* abundance correlated with nausea severity.
El Jurdi et al. (2019) [51]	What are the longitudinal dynamics of bacteriome and mycobiome recovery post-ASCT?	Prospective non-randomized pilot study	N = 15 (ASCT = 15, Allo-HCT = 0)	16S rRNA sequencing for bacteriome and mycobiome	Correlation of bacteriome/mycobiome changes with clinical outcomes. Limited statistical rigour for small sample size.	2b	Newcastle-Ottawa	Focus on both bacteriome and mycobiome dynamics; clinical applicability.	Small sample size; limited follow-up period.	Bacteriome recovers within 1 month; mycobiome diversity remains disrupted longer.
Shah et al. (2022) [46]	Can dietary interventions and stool butyrate levels predict sustained MRD negativity in multiple myeloma?	Cross-sectional observational study	N = 48 (ASCT = 48, Allo-HCT = 0)	16S rRNA sequencing and dietary data integration	Logistic regression for microbiota markers and MRD status; inclusion of dietary data for broader relevance.	2b	CEBM	Novel integration of dietary factors and microbiota composition.	Cross-sectional design; observational correlations lack mechanistic insights.	Plant-based diets and higher stool butyrate linked to sustained MRD negativity in MM. Positive: Butyrate-producing taxa (*Eubacterium hallii* and *Faecalibacterium prausnitzii*) associated with sustained MRD negativity and better outcomes.
Schwabkey et al. (2022) [48]	What bacterial taxa are associated with febrile neutropenia during severe neutropenia post-ASCT?	Single-centre observational study with preclinical models	N = 119 (ASCT = 63, Allo-HCT = 56)	16S rRNA sequencing and mouse model validation	Permutational MANOVA for beta diversity; mouse models validated human data findings.	2a	Newcastle-Ottawa	Combined human and preclinical models; robust microbiota-metabolite analyses.	Observational study design for human data; potential overinterpretation of mouse results.	*A. muciniphila* abundance correlates with febrile neutropenia in post-HCT patients.

**Table 2 microorganisms-13-02302-t002:** Summary of taxa, key roles, and clinical outcomes.

Bacterial Taxa	Key Role	Impact on Clinical Outcomes	Proposed Mechanism
*Eubacterium hallii*	Butyrate production	Improved PFS, MRD negativity	SCFA production, T-reg induction, gut barrier integrity.
*Akkermansia muciniphila*	Mucin degradation	Higher infection risk, febrile neutropenia	Excess mucin degradation weakens gut barrier, increasing inflammation.
*Faecalibacterium prausnitzii*	Anti-inflammatory, butyrate producer	Poor outcomes with depletion	SCFA production, anti-inflammatory effects, gut barrier protection.
*Blautia* spp.	SCFA production	Better PFS and OS	Supports epithelial health, anti-inflammatory signalling.
*Klebsiella* spp.	Nitrogen recycling	Tumour progression in MM	Promotes glutamine biosynthesis, enhancing tumour survival.
*Ruminococcus* spp.	SCFA production	Suboptimal therapeutic responses when depleted	Supports epithelial integrity and anti-inflammatory signalling.
*Prevotella* spp.	Pro-inflammatory	Tumour-promoting inflammation	IL-17-mediated inflammatory pathways linked to disease progression.

PFS = Progression Free Survival, OS = Overall Survival, SCFA = Short Chain Fatty Acids, MM = Multiple Myeloma.

## 4. Discussion

The relationship between gut microbiome (GM) composition and outcomes following autologous stem cell transplantation (ASCT) is an emerging area of research. While the impact of GM on allogeneic transplantation outcomes has been extensively studied, similar investigations in ASCT remain limited. This systematic review synthesises the findings of 11 studies to evaluate the associations between GM diversity and ASCT outcomes. The discussion highlights key mechanisms, identifies challenges in interpreting existing evidence, and explores future directions.

### 4.1. Challenges in Synthesising ASCT Studies

The heterogeneity of patient populations and study methodologies poses significant challenges to synthesising findings across ASCT studies. Variability in underlying diseases (e.g., multiple myeloma, lymphoma), conditioning regimens, dietary habits, and antibiotic exposures contributes to differences in GM composition and its relationship with clinical outcomes. For instance, Khan et al. [42] and Schwabkey et al. [48] both identified associations between GM diversity and survival outcomes, yet their results are influenced by differences in patient demographics, antibiotic use, and study designs.

The bidirectional confounding effect of chemotherapy, antibiotic exposure, and the underlying malignancy further complicates causal inference. A healthy GM promotes immune recovery and reduces infection risks, initiating a positive feedback loop that supports microbial diversity. In contrast, dysbiosis often leads to infections, increased antibiotic use, and further disruption of the microbiota, creating a self-perpetuating cycle. Antibiotics, while essential for managing infections, emerge as a major driver of GM dysbiosis in ASCT, compounding the difficulty of isolating the microbiome’s specific contributions to clinical outcomes [51]. These interdependencies underscore the need for longitudinal designs and advanced analytical techniques, such as machine learning, to better understand GM dynamics over time [49].

### 4.2. Associations Between the GM and ASCT Outcomes

#### 4.2.1. Beneficial Bacteria

*Faecalibacterium prausnitzii*: This butyrate-producing bacterium has been consistently associated with improved ASCT outcomes. Shah et al. [46] demonstrated that higher relative abundance of *F. prausnitzii* was linked to sustained minimal residual disease (MRD) negativity in multiple myeloma patients. Similarly, Khan et al. [42] found that increased alpha diversity, often driven by beneficial taxa like *F. prausnitzii*, was associated with a 48% reduction in mortality risk (HR: 0.52; *p* = 0.003). Butyrate produced by *F. prausnitzii* enhances gut barrier integrity and promotes regulatory T-cell (T-reg) differentiation, thereby reducing inflammation and supporting immune recovery [49].

*Blautia* spp.: A higher abundance of *Blautia* spp. correlates with reduced gastrointestinal toxicities, such as mucositis and diarrhoea, as observed by Laheij et al. [49]. Butyrate produced by these bacteria strengthens epithelial tight junctions, preventing endotoxin translocation and systemic inflammation. Additionally, D’Angelo et al. [45] reported that a high-fibre diet increased *Blautia* abundance, linking dietary interventions to improved microbiota profiles and treatment outcomes [50].

#### 4.2.2. Pathogenic Bacteria

*Klebsiella* spp.: Jian et al. [52] highlighted the pathogenic role of *Klebsiella* spp., a nitrogen-recycling bacterium that exacerbates tumour progression by promoting glutamine biosynthesis. The study’s mechanistic insights underscore the potential for targeting *Klebsiella* in microbiota-based interventions to mitigate disease progression in multiple myeloma [50].

*Enterococcus* spp.: Khan et al. [42] associated a higher abundance of *Enterococcus* spp. with increased mortality risk (HR: 5.76; *p* = 0.011). The ability of *Enterococcus* to disrupt gut barrier integrity and trigger systemic inflammation highlights the need for careful monitoring of this taxon during ASCT [50].

*Proteobacteria:* Elevated levels of *Proteobacteria* have been linked to gut barrier dysfunction and increased inflammation. Schwabkey et al. [48] found that there is an overrepresentation of *Akkermansia muciniphila* and *Bacteroides* spp. during neutropenia, a correlation was observed with febrile episodes, further emphasising the importance of maintaining a balanced GM [49].

### 4.3. Mechanistic Pathways

SCFA Production: Beneficial taxa such as *Faecalibacterium prausnitzii* and *Blautia* spp. ferment dietary fibre into short-chain fatty acids (SCFAs), including butyrate, which enhances gut barrier integrity by supporting tight junctions and colonocyte energy metabolism while modulating immune responses through T-reg differentiation. These mechanisms contribute to reduced inflammatory complications and improved post ASCT survival [49].

Pathogen Resistance: Protective taxa, including *Bacilli* and *Erysipelotrichales*, enhance colonisation resistance by competing with pathogens for resources and producing antimicrobial peptides. Schwabkey et al. [48] observed that these taxa were enriched in patients who remained afebrile, suggesting their role in reducing infection risk during periods of neutropenia [49].

Gut Barrier Integrity: *Akkermansia muciniphila* and *Blautia* spp. contribute to gut barrier health by maintaining mucosal thickness and epithelial integrity. However, dysbiosis-driven overrepresentation of *Akkermansia* has been associated with febrile neutropenia, highlighting the complex role of this bacterium in ASCT [49].

### 4.4. Probiotic Interventions in ASCT

Probiotics have shown mixed results in modulating the GM and improving ASCT outcomes. While multi-strain probiotics have been associated with reduced bloodstream infections and faster recovery of microbiota diversity [50], challenges such as poor colonisation in dysbiotic environments limit their efficacy. Comparatively, faecal microbiota transplantation (FMT) offers broader restoration of microbial diversity but is more invasive. Dietary interventions, including high-fibre supplementation, provide a non-invasive and sustainable approach to improving GM composition, as demonstrated by D’Angelo et al. [45]. Combining probiotics with prebiotics or personalised dietary strategies may further enhance their efficacy.

### 4.5. Critical Evaluation of Statistical Models

The statistical methodologies employed across studies investigating the GM in ASCT reveal both strengths and weaknesses. While some studies effectively leveraged advanced modelling to account for confounding variables and capture longitudinal trends, others suffered from limitations such as small sample sizes, inadequate confounder adjustment, and a lack of mechanistic depth. This section compares and contrasts statistical approaches across the reviewed studies.

### 4.6. Strengths in Statistical Approaches

Survival analyses were a common approach in several studies, with notable differences in their rigour and outcomes. Khan et al. [42] used multivariable Cox regression to explore the relationship between GM diversity and survival outcomes, reporting a statistically significant association between higher alpha diversity and improved overall survival (OS) and progression-free survival (PFS). This study adjusted for critical potential confounders, including age, disease type, and conditioning regimen, to ensure robust findings. However, it did not explicitly test the proportional hazards assumption, which could potentially limit the reliability of the conclusions. Similarly, Schwabkey et al. [48] employed time-to-event analyses to investigate the association between specific taxa, such as *Akkermansia muciniphila* and *Bacilli*, and febrile neutropenia. While this study provided valuable temporal insights, it did not differentiate between ASCT and allo-HSCT patients, thereby reducing the specificity of its findings. When compared, both studies offered complementary insights into survival outcomes, with Khan’s study benefiting from a more robust adjustment for confounders, whereas Schwabkey’s analysis provided greater granularity regarding temporal changes in microbiota.

Some studies employed longitudinal designs to capture dynamic GM changes over time, offering critical insights into temporal relationships. Laheij et al. [49] used mixed-effects models to investigate the protective role of *Bacteroidetes* against gastrointestinal toxicity. This approach allowed the study to account for intra-individual variability over time, demonstrating significant findings despite a small sample size of 51 patients. In contrast, Bansal et al. [45] compared longitudinal microbiota changes between ASCT and allo-HSCT patients, observing faster recovery of GM diversity in the former group. However, Bansal’s analysis lacked the temporal modelling depth of Laheij, as it did not explore dynamic interactions between microbiota changes and clinical outcomes. Comparing these studies highlights the strength of mixed-effects models in capturing temporal data, as seen in Laheij’s study, while also underscoring the need for such approaches in studies like Bansal’s to enhance their findings.

Mechanistic integration was a key feature of studies that aimed to link GM composition to functional pathways and clinical outcomes. Jian et al. [52] combined human data with preclinical models to demonstrate how *Klebsiella* spp. promotes tumour progression through glutamine biosynthesis. This integrative approach enhanced the biological plausibility of the findings but was constrained by a small cohort size of 19 patients. Similarly, Shah et al. [47] investigated the association between butyrate-producing taxa, such as *Faecalibacterium prausnitzii* and *Eubacterium hallii*, and the negativity of minimal residual disease (MRD) in patients with multiple myeloma. While Shah’s study provided strong clinical correlations, it lacked the functional depth seen in Jian’s work. Together, these studies demonstrate the importance of integrating mechanistic insights with clinical observations, highlighting the need for larger cohorts to validate these findings.

### 4.7. Limitations in Statistical Approaches

We acknowledge several significant limitations of this analysis, including small sample sizes, heterogeneity in reporting across studies, inconsistent reporting of results and confounder control, along with evolving mechanistic insights into the processing linking GM diversity and meaningful clinical outcomes.

Small sample sizes significantly constrained the reliability of findings in several studies. Kusakabe et al. [44] reported a striking 66.7% two-year mortality rate in patients with dysbiosis, but the small cohort size reduced the generality of this observation. Similarly, El Jurdi et al. [51] examined the relationship between GM composition and gastrointestinal toxicities but included fewer than 30 participants, further limiting the generalisability of the results. Both studies highlight the challenge of drawing robust conclusions from underpowered datasets, emphasising the need for larger, multicentre trials to confirm these findings.

Confounder control was inconsistent across studies, with some adjusting for key variables while others did not. D’Angelo et al. [45] explored the impact of dietary interventions on GM composition and ASCT outcomes. Still, they did not adequately adjust for potential confounders such as antibiotic use or disease severity, which may have biassed the results. Similarly, Pianko et al. [47] linked the abundance of *Eubacterium hallii* to MRD negativity but lacked detailed data on dietary patterns or antibiotic exposure. In contrast, Khan et al. [42] demonstrated a more rigorous approach to confounder adjustment, incorporating multiple variables such as age and conditioning regimens into the analyses. Comparing these studies underscores the critical role of comprehensive confounder control in ensuring the reliability of microbiota research.

Effect sizes and confidence intervals were not consistently reported across studies, reducing the interpretability of their findings. Khan et al. [42] provided hazard ratios and confidence intervals for their survival analyses, enhancing the robustness of their conclusions. In contrast, Laheij et al. [49] reported significant *p*-values but did not include effect sizes or confidence intervals, limiting the clarity of its results. This inconsistency across studies highlights the importance of comprehensive statistical reporting to facilitate meaningful comparisons and meta-analyses.

### 4.8. Comparisons Across Study Designs

Temporal analyses offered richer insights into microbiota dynamics compared to static approaches. Laheij et al. [49] excelled in capturing changes in GM composition over time, linking temporal shifts in *Bacteroidetes* abundance to gastrointestinal toxicity outcomes. In contrast, Shah et al. [46] provided a static snapshot of GM composition, identifying associations between beneficial taxa and MRD negativity without exploring how these relationships evolved. This comparison highlights the added value of temporal designs in uncovering dynamic microbiota-host interactions.

Mechanistic studies provided deeper biological insights compared to purely observational approaches. Jian et al. [52] used functional metagenomics and preclinical models to establish a mechanistic link between *Klebsiella* spp. and tumour progression, offering actionable insights into potential therapeutic targets. In contrast, observational studies, such as those by Schwabkey et al. [48], identified associations between specific taxa and clinical outcomes but lacked mechanistic depth. Together, these studies underscore the complementary roles of observational and mechanistic research in advancing our understanding of the GM’s role in ASCT.

## 5. Conclusions and Future Directions

This systematic review highlights our current understanding of the gut microbiome’s (GM) role in autologous stem cell transplantation (ASCT). Key findings include the beneficial role of butyrate-producing organisms such as *Faecalibacterium*, which may be enhanced through targeted probiotic and dietary strategies avoiding indiscriminate use of antimicrobials known to have specific toxicity. Similarly, the detrimental effects of *Klebsiella* and *Enterococcus* species underscore the need for careful monitoring of these taxa and early intervention in the event of clinical infection.

An important initial step to improving the quality of evidence in this area is expanding patient cohorts. Larger sample sizes will increase statistical power and precision and improve the reliability of findings. Multicentre, collaborative studies employing harmonised protocols are crucial for ensuring diverse patient populations, which can yield more robust results. Equally important is the standardisation of methodologies across studies. Adopting uniform approaches to microbiota analysis, such as favouring shotgun metagenomics over 16S rRNA sequencing, will provide deeper functional insights. Rigorous control of confounding factors, including antibiotic use, dietary variations, and disease severity, is necessary to isolate the specific role of the GM in ASCT outcomes and distinguish true correlation from bystander effects.

Additional future priorities for the field include the use of in vitro models, including organoids, cell lines, and longitudinal sampling to capture the dynamic changes in the GM pre- and post-transplantation. Such designs will allow researchers to identify causal relationships and temporal trends with greater precision. Mechanistic studies using non-human models, such as germ-free and humanised mouse models, represent another critical avenue for future research. These models offer controlled environments in which microbiota–host interactions can be explored. For example, researchers can test hypotheses about the role of specific bacterial taxa (i.e., *Faecalibacterium prausnitzii*), in immune regulation. Additionally, these models could help elucidate the functional impact of dysbiosis on engraftment, systemic inflammation, and infection, as well as assess the effects of therapeutic interventions, such as probiotics or faecal microbiota transplantation (FMT), in a controlled setting. By combining these approaches, future research can move beyond observational associations to define mechanistic pathways and identify actionable therapeutic targets. This integrated strategy will help translate microbiota research into clinical interventions, ultimately improving outcomes for ASCT patients.

Important limitations of the existing literature include the observational nature of the evidence and the paucity of standardised, multicenter, longitudinal studies. The statistical methodologies used in ASCT studies vary widely in their rigour and scope, with each approach offering strengths and limitations. Studies by Khan et al. [42] and Laheij et al. [49] employed robust modelling techniques to explore survival outcomes and temporal changes in microbiota, respectively. In contrast, smaller studies, such as those by Kusakabe et al. [44] and El Jurdi et al. [51], provided exploratory insights but were limited by underpowered designs. The inconsistent control of confounders and incomplete reporting of effect sizes further constrain the interpretability of findings across the field. Future research must prioritise the integration of advanced statistical approaches, larger cohorts, and functional data to enhance the reliability and depth of microbiota research in ASCT.

This review highlights multiple observed associations between GM composition and clinical outcomes in ASCT, including relationships between specific bacterial taxa, gut integrity, immune recovery, and infection risk. Although promising, the evidence remains observational, with limited mechanistic insights or demonstrated effects. Preclinical studies, such as germ-free mouse models and in vitro systems, can complement human data to generate hypotheses, define mechanistic explanations, and explore potential therapeutic interventions.

By advancing the rigour of observational research and integrating preclinical models, the field can progress toward a more comprehensive understanding of the microbiome’s role in ASCT and its therapeutic potential.

## Figures and Tables

**Figure 1 microorganisms-13-02302-f001:**
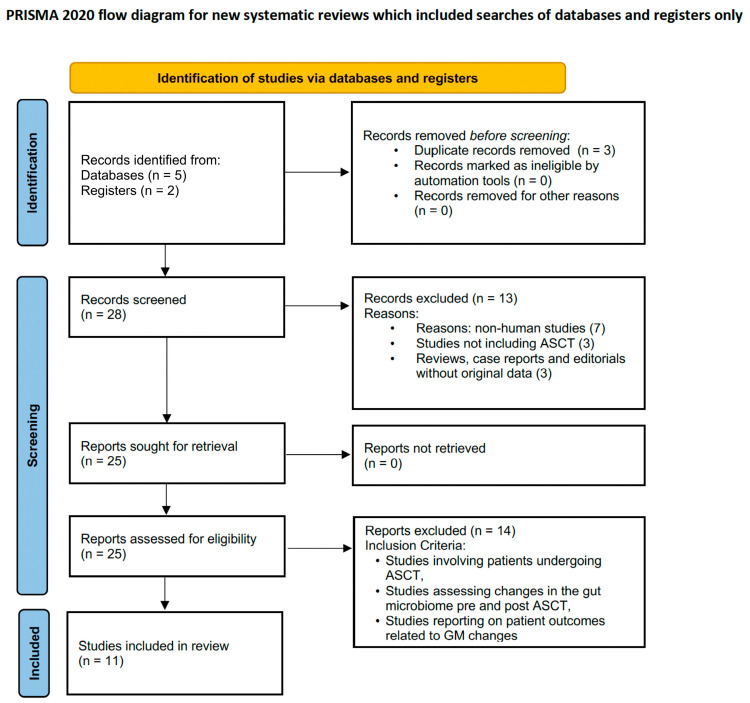
Flow diagram of the systematic review process [43].

## Data Availability

No new data were created or analyzed in this study. Data sharing is not applicable to this article.

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
