# Peer review of "Associations Between the Gut Microbiome and Outcomes in Autologous Stem Cell Transplantation: A Systematic Review"

_microorganisms, 2025, doi:10.3390/microorganisms13102302_

Round 1
Reviewer 1 Report
Comments and Suggestions for Authors
The manuscript addresses an important and timely topic, exploring the relationship between the gut microbiome and outcomes in autologous stem cell transplantation (ASCT). The topic has strong clinical and translational relevance, and the review makes a valuable attempt at synthesizing available evidence. However, several methodological, structural, and referencing issues need to be addressed before the manuscript can be considered for publication.
Decision: minor revisions
Abstract
- Missing structured presentation (Background, Methods, Results, Conclusion).
- Overstates conclusions; should better reflect study limitations.
- Inconsistency: 11 vs. 12 studies included. Needs correction.
Introduction
The introduction relies heavily on older references, while more recent studies on microbiome and ASCT are available. I recommend integrating the following to strengthen the background and discussion:
- https://doi.org/10.1007/s12035-025-04846-0
- https://doi.org/10.3389/fbioe.2025.1571066
- https://doi.org/10.1002/hsr2.2036
- https://doi.org/10.1002/iid3.70189
- https://doi.org/10.3390/immuno4040026
- https://doi.org/10.5662/wjm.v15.i2.92592
Materials and Methods
- Lacks detail on search terms, time frame, and risk of bias assessment.
- Inconsistency in study numbers (11 vs. 12). Clarify and align text, PRISMA flowchart, and results.
Results
- Inconsistent reporting of included studies.
- Some studies are summarized with numerical detail (HR, %), while others are only qualitatively described. A uniform approach is needed.
Discussion
- Reiterates findings without sufficient critical appraisal.
- Needs a more balanced discussion of limitations: small sample sizes, heterogeneity, confounders, and limited mechanistic insights.
Conclusion
- Too general and advocacy-driven.
- Should emphasize the observational nature of the evidence and the need for standardized, multicenter, longitudinal studies.
- Future directions (already partially covered) could be summarized more concisely.
Minor Comments
- Grammar and style need polishing for clarity and flow.
- Ensure consistency in bacterial taxa names (italicize appropriately, e.g., Faecalibacterium prausnitzii).
- Some abbreviations (e.g., MRD, GVHD) should be defined at first mention in each section.
Author Response
Response to Reviewer 1:
We thank the reviewer for their overall positive and insightful review of our manuscript. We have addressed the points raised, which we believe have significantly strengthened the article. Please find below a point-by-point response to each comment.
- Abstract: Missing structured presentation (Background, Methods, Results, Conclusion). Overstates conclusions; should better reflect study limitations. Inconsistency: 11 vs. 12 studies included. Needs correction.
We thank the reviewer for this comment and agree that it is important to acknowledge study limitations appropriately. The abstract has been restructured to reflect this. 11 studies were included in this review, and the PRISMA flowchart has been corrected to reflect this.
- Introduction: The introduction relies heavily on older references, while more recent studies on microbiome and ASCT are available. I recommend integrating the following to strengthen the background and discussion:
We appreciate this comment from the reviewer and agree with the importance of providing an up-to-date introduction. This has been accordingly revised, incorporating the most relevant of the reviewer’s suggested recent references regarding the broader role of the gut microbiome in health and disease. This is indeed important data to present. Given the relatively limited space in this publication we are only able to briefly discuss this work.
- Materials and Methods: Lacks detail on search terms, time frame, and risk of bias assessment.
Inconsistency in study numbers (11 vs. 12). Clarify and align text, PRISMA flowchart, and results.
This section has been reviewed, and these inconsistencies have been corrected. The MeSH search terms are listed in lines 194-195. Time frames have been added. Final search result numbers have been collected and the PRISMA flowchart (Figure 1) has been updated to align with the rest of the paper. The Newcastle-Ottawa scale was used for risk-of-bias assessment and this has been clarified in the paper.
- Results: Inconsistent reporting of studies. Some studies are summarized with numerical detail (HR, %), while others are only qualitatively described. A uniform approach is needed.
Wherever possible, the inconsistencies in the reporting of studies have been addressed, within the limitations of how these data have been presented in the primary literature. This issue has been highlighted in Section 4.6 of the Discussion, where the following statement has been added:
4.7. Limitations in Statistical Approaches
‘We acknowledge several significant limitations of this analysis, including small sample sizes, heterogeneity in reporting across studies, inconsistent reporting of results and confounder control, along with evolving mechanistic insights into the processing linking GM diversity and meaningful clinical outcomes.’
- Discussion: Reiterates findings without sufficient critical appraisal. Needs a more balanced discussion of limitations: small sample sizes, heterogeneity, confounders, and limited mechanistic insights.
This section has been reviewed and these discussion of these limitations has been appropriately highlighted. For example in Section 4.7, we say that ‘small sample sizes significantly constrained the reliability of findings in several studies. Kusakabe et al. [46] reported a striking 66.7% two-year mortality rate in patients with dysbiosis, but the small cohort size reduced the generality of this observation’ and then further say ‘Confounder control was inconsistent across studies, with some adjusting for key variables while others did not. D’Angelo et al. [47] explored the impact of dietary interventions on GM composition and ASCT outcomes. Still, they did not adequately adjust for potential confounders such as antibiotic use or disease severity, which may have biased the results. Similarly, Pianko et al. [49] linked the abundance of Eubacterium hallii to MRD negativity but lacked detailed data on dietary patterns or antibiotic exposure. In contrast, Khan et al. [44] demonstrated a more rigorous approach to confounder adjustment, incorporating multiple variables such as age and conditioning regimens into the analyses.’
- Conclusion: Too general and advocacy-driven. Should emphasize the observational nature of the evidence and the need for standardized, multicenter, longitudinal studies.Future directions (already partially covered) could be summarized more concisely.
We thank the reviewer for this excellent point which we have incorporated to refine and sharpen the focus of the conclusion.
- Minor Comments. Grammar and style need polishing for clarity and flow. Ensure consistency in bacterial taxa names (italicize appropriately, e.g., Faecalibacterium prausnitzii).Some abbreviations (e.g., MRD, GVHD) should be defined at first mention in each section.
We thank the reviewer for identifying these improvements, which have been addressed. For example, Klebsiella has been appropriately italicised in Table 1.
Reviewer 2 Report
Comments and Suggestions for Authors
This manuscript presents a systematic review examining the associations between gut microbiota and clinical outcomes in autologous stem cell transplantation (ASCT). The authors analysed 11 studies, including both observational studies and clinical trials. The main findings show a consistent presence of beneficial bacteria (Bacteroidetes, Firmicutes, Faecalibacterium prausnitzii) and an increase in pathogenic bacteria (Escherichia coli, Enterococcus spp., Klebsiella spp.) post-transplantation. The review highlights the importance of microbial diversity in improving ASCT outcomes and identifies gaps in the current literature.
- The flow chart shows that 12 studies were included in the final review, but the abstract and text mention 11 studies. This discrepancy needs to be resolved.
- A systematic assessment of the risk of bias for the included studies is not provided, which is standard for systematic reviews.
- The search strategy does not specify the cut-off dates or time limitations applied.
- The discussion should more clearly include the practical implications for the clinical management of ASCT patients.
- The recommendations for future research (section 5) are too broad. More specific priorities and realistic timelines are needed.
Author Response
Response to Reviewer 2
We thank the reviewer for their critical appraisal of our manuscript. We have addressed the points raised which we believe have significantly strengthened the article and clarified its key messages. Please find below a point-by-point response to each comment.
- The flow chart shows that 12 studies were included in the final review, but the abstract and text mention 11 studies. This discrepancy needs to be resolved.
We appreciate the reviewer pointing out this discrepancy, which has been accordingly addressed. The correct number is 11 studies.
- A systematic assessment of the risk of bias for the included studies is not provided, which is standard for systematic reviews.
Assessment for risk of bias was conducted using the accepted Newcastle-Ottawa Scale. This has been clarified in the manuscript.
- The search strategy does not specify the cut-off dates or time limitations applied.
We apologise for this oversight. This has been corrected in Section 2, paragraph 2:
The most recent search was conducted on 27th July, 2025.
- The discussion should be more clearly framed to include the practical implications for the clinical management of ASCT patients.
We thank the reviewer for emphasising the implications for patient management, which is a key focus of this review. An introductory paragraph has been added to the discussion to highlight these based on the findings of this review.
- The recommendations for future research (section 5) are too broad. More specific priorities and realistic timelines are needed.
We agree that the future priorities should be presented in a more targeted and concise way. This has been addressed in the manuscript.
We thank the reviewers again for their efforts.